# Congenital Viral Infection Risk: The Role of Parvovirus B19 and Cytomegalovirus Molecular Genetic Testing

**DOI:** 10.3390/ijms27010427

**Published:** 2025-12-31

**Authors:** Stefka Krumova, Ivelina Trifonova, Mariela Hristova-Savova, Lora Veleva, Radostina Stefanova, Petia Genova-Kalou, Petya Chaveeva, Vasil Kalev, Tanya Tilkova, Tsvetoslav Vassilev, Ivanka Dimova

**Affiliations:** 1National Reference Laboratory “Measles, Mumps, Rubella”, Department of Virology, National Center of Infectious and Parasitic Diseases, 1000 Sofia, Bulgaria; loraveleva9@gmail.com (L.V.); rss_94@abv.bg (R.S.); 2Center of Competence “ImmunoPathogen”, 1000 Sofia, Bulgaria; 3National Reference Laboratory “NRL of Influenza and Acute Respiratory Diseases”, Department of Virology, National Center of Infectious and Parasitic Diseases, 1000 Sofia, Bulgaria; trifonova.ivelina@abv.bg; 4National Reference Laboratory “Herpes and Oncogenic Viruses”, Department of Virology, National Center of Infectious and Parasitic Diseases, 1000 Sofia, Bulgaria; eli.savova@abv.bg; 5Medical Complex “Dr. Shterev”, 1000 Sofia, Bulgaria; chaveevapetya@gmail.com (P.C.); vasil.kalev@gmail.com (V.K.); tania_tilkova@icloud.com (T.T.); tgvassilev@gmail.com (T.V.); ivanka.i.dimova@gmail.com (I.D.); 6National Reference Laboratory “Cell Cultures and Rickettsia”, Department of Virology, National Center of Infectious and Parasitic Diseases, 1000 Sofia, Bulgaria; petia.d.genova@abv.bg; 7Faculty of Medicine, Medical University of Sofia, 1000 Sofia, Bulgaria

**Keywords:** pregnancy, B19V, CMV, congenital infection, antenatal screening

## Abstract

Parvovirus B19 and cytomegalovirus are significant causes of congenital infections that can lead to adverse pregnancy outcomes. The present study aimed to investigate the infection of B19V and CMV in pregnant women with fetal anemia, effusions and intrauterine growth restriction and determine the utility of routine laboratory screening in pregnancy follow-up. Thirteen women with such pathological pregnancy complications attending an antenatal clinic from April 2024 to March 2025 were tested. Three types of clinical material were examined: maternal blood, amniotic fluid and umbilical cord serum. Participants underwent molecular and serological testing for both B19V and CMV. Demographic data, obstetric histories, and pregnancy outcomes were recorded and analyzed. Our results indicate that three participants showed evidence of either current infection with CMV and seven with B19V. Pregnant women with active infections required further follow-up and fetal surveillance. A stillbirth was reported in one woman with CMV infection. For seven samples that tested positive for B19V DNA, viral sequences were obtained and clustered with genotype 1a reference strains. The findings of this study highlight the significant contribution of B19V and CMV infections during pregnancy, particularly in cases complicated by fetal anemia, effusions, and intrauterine growth restriction.

## 1. Introduction

Viral infections during pregnancy are a significant cause of severe complications and mortality for both mothers and fetuses worldwide, even in high-income countries. Infectious agents can reach the fetus in several ways, transplacentally, perinatally (through vaginal secretions or blood), or postnatally (via breast milk), and can adversely affect the health of the pregnant woman, the fetus, or the newborn [1,2,3,4]. Generally, the risk of neonatal infections is inversely proportional to the gestational age at acquisition [1,5].

Parvovirus B19 (B19V) and cytomegalovirus (CMV) are pathogens that may be transmitted prenatally through the transplacental route or perinatally through blood or vaginal secretions and can lead to serious complications, including anemia, spontaneous abortion, premature birth, and stillbirth. Both viruses are present worldwide, and there are currently no specific vaccines available for either. Additionally, B19V re-emerged across Europe and Bulgaria in 2024, raising concerns about vertical transmission and neonatal morbidity [6,7].

B19V is a single-stranded DNA virus from the *Parvoviridae* family and *Erythrovirus* genus, primarily causing erythema infectiosum, commonly called fifth disease. While a significant percentage of adults may experience asymptomatic infections, the consequences of B19V infection during pregnancy can be severe, including fetal anemia, hydrops fetalis and intrauterine fetal death [1]. B19V particles can cross the placenta and inhibit fetal erythropoiesis, leading to aplastic crisis and subsequent congestive anemia, hypoxia, and heart failure [1,8,9,10]. The probability of experiencing these complications after 20 weeks of gestation is approximately 2.3% [11,12]. Research indicates that between 50% and 65% of women of reproductive age have developed immunity to this virus [8,9,10]. During epidemic outbreaks, it is estimated that 20% to 30% of seronegative women (those who have not previously been exposed to the virus) will contract the infection.

CMV is a double-stranded DNA virus that belongs to the family *Orthoherpesviridae*, (subfamily *Betaherpesvirinae*, genus *Cytomegalovirus*), and is one of the most common congenital viral infections, presenting with a range of clinical manifestations. In most cases, CMV infection is asymptomatic (90%), but it can occasionally present as a mild febrile illness with nonspecific symptoms such as fatigue, myalgia, rhinitis, pharyngitis, and headache [13]. Among pregnant individuals, CMV has a seroprevalence of 42.3–68.3% in developed countries. In addition to the main clinical manifestations of the disease, the virus can also lead to anemia and alterations in serum iron metabolism [14,15,16]. The risk of transmission of the infection to the fetus is highest during primary infection, estimated at 30–40% [17].

The gold standard in diagnosing fetal anemia is the measurement of the middle cerebral artery peak systolic velocity (MCA-PSV) during 18–35 weeks’ gestation. MCA-PSV Doppler is a standard (routine-in-risk) test in specialized prenatal/maternal–fetal medicine units, primarily for pregnancies with known or suspected risk of fetal anemia, such as alloantibodies in the mother, maternal viral infection, abnormal fetal heart rate, etc.

Laboratory diagnosis and established methods for confirming B19V and CMV infections rely on serological and molecular techniques. Testing for specific B19V and CMV IgM antibodies, combined with the presence of viral DNA, provides definitive evidence of acute infection, which is particularly important for monitoring cases of pathological pregnancy [17]. To study the dynamics of viral evolution and their epidemiological distribution, next-generation sequencing (NGS) techniques are increasingly utilized.

Taking all of this into consideration, the present study aimed to investigate the infection of B19V and CMV in pregnant women with fetal anemia, effusions and intrauterine growth restriction, evaluate the utility of routine molecular and serological viral screening, and investigate its role in pregnancy follow-up.

## 2. Results

Over a period of twelve months, 13 pregnant women with fetal anemia, effusions, and intrauterine growth restriction were screened for two of the most common viral agents that can cause congenital infections if transmitted from a pregnant woman to her baby: B19V and CMV.

### 2.1. B19V Infection

In 7 patients (54%), B19V infection was confirmed. The main reported clinical complications included fetal anemia, ascites, and hydrops fetalis. All confirmed patients are in the second trimester of pregnancy. The virus was detected in all clinical specimens using PCR, while serological methods (ELISA for IgM and IgG) confirmed the infection in four cases. In four instances, follow-up clinical samples were collected from both the fetus in utero and the mother. Notably, positivity was detected at a very early cycle, with cycle threshold (Ct) values ranging from 13 to 19 in samples taken in utero. A mostly favorable outcome of the pregnancies was reported (see Table 1).

During pregnancy, when B19V infection was diagnosed, an intrauterine blood transfusion was performed. This procedure successfully controlled congenital anemia and ensured the normal development of the fetus in the monitored women. Normal fetal development and live births were observed in five of the patients studied who were confirmed to have B19V infection. In the remaining two patients, control of fetal anemia was also noted; however, they were not followed up because they continued their care in other gynecological offices.

Whole-genome sequencing was performed in seven B19V-positive samples using the NGS method. All Bulgarian sequences were associated with genotype 1a by phylogenetic analysis. Therefore, genotype 1a of B19V was detected in clinical materials from both the mother and the fetus. The Bulgarian sequences showed the closest phylogenetic similarity to those from the Netherlands and Italy in 2024 (see Figure 1).

### 2.2. CMV Infection

In the study, 3 out of the 13 pregnant women (23%) were confirmed with CMV infection. The ages of these patients were 19, 28 and 32. The most frequently reported complications included fetal growth restriction, as evidenced by changes in the brain, heart, and intestines. CMV infection was validated through real-time PCR in clinical samples, including amniotic fluid and umbilical cord serum. Unfortunately, one of these patients experienced stillbirth during the pregnancy (see Table 2). In one patient, who was pregnant in the third trimester (30 weeks gestation), CMV IgM and IgG antibodies were detected by serological methods (ELISA), but she refused to have an amniocentesis and CMV infection was not confirmed by PCR.

No sequencing was performed on the CMV-positive samples since all had a Ct value above 30.

In two of the thirteen (2/13, 15%) monitored pregnant women, no B19V or CMV infection was detected, and the cause of the developed pathology is likely due to another factor.

## 3. Discussion

The present study underscores the clinical relevance of maternal molecular genetic screening for B19V and CMV as key determinants of congenital infection risk. B19V is not regarded as a teratogenic agent that affects embryogenesis during the first 8–10 weeks of gestation; therefore, it is not an indication for pregnancy termination. However, it is recognized as a human pathogen that can infect the placenta [16]. Reports suggest that B19V infection during pregnancy occurs in about 1–5% of women, with the estimated rate of vertical transmission during maternal infection ranging from 17% to 33% [18]. Data in the literature suggest that when a seronegative mother becomes infected, the fetus has a favorable outcome in 85% of cases [19]. The effects of B19V infection on the fetus can vary widely, ranging from asymptomatic carrier status to spontaneous abortion, hydrops fetalis, congenital anemia, and intrauterine fetal death [20]. Since early 2024, several European countries have reported an increase in B19V infections, particularly among pregnant women and children. This trend emphasizes the need for enhanced epidemiological surveillance, especially given the limitations of routine monitoring and the potential impacts of COVID-19-related immunity gaps in the post-pandemic period [6,7,21]. In Bulgaria, where an overall increase in B19V incidence was noted during this study, thirteen women with pathological pregnancies were screened. Our results indicated that all confirmed B19V patients experienced favorable pregnancy outcomes following primary B19V infection. However, this was associated with a heightened risk of vertical transmission and reported conditions such as fetal anemia, hydrops, or ascites. All women with B19V in our study were diagnosed with anemia, which was managed during their pregnancies. Intravenous blood transfusions were performed to manage congenital anemia, leading to normal fetal development [6]. Targeted serological testing during B19V outbreaks, combined with weekly Doppler surveillance of fetuses with confirmed maternal infection, remains the most evidence-based approach to preventing hydrops and optimizing the timing of intrauterine transfusions [6,22]. B19V was successfully isolated from maternal and intrauterine fetal blood samples, with serum PCR assays detecting high levels of viremia.

This study marks the first time that sequencing of B19V from clinical samples taken in utero has been conducted in the country. The comprehensive NGS analysis confirms the presence of the B19V 1a genotype in the examined cohort. Similar studies from 2024 indicate that genotype 1a is the dominant strain in Europe among various groups of infected individuals [7,23,24]. Due to the uniform genetic affiliation of the B19V genetic sequences we isolated, we did not detect any evolutionary advantage in terms of genotype and clinical complications. Previous studies on the circulation of B19V in Bulgaria, particularly among patients with fever rash syndrome and those with hematological disorders, have confirmed that genotype 1a continues to persist within the country’s territory [25,26].

CMV can be transmitted vertically at any stage of pregnancy. The most severe effects on the fetus are associated with infections that occur during the first trimester, with the severity of the disease decreasing as gestational age increases. The highest risk of transmitting the infection to the fetus occurs during primary infection, and in 0.2% to 2.5% of seropositive pregnant women, premature birth may happen [17]. Statistics indicate that 30% of newborns with severe congenital CMV infection do not survive [16]. Among those who survive, more than half will ultimately develop neurological complications. Congenital CMV infection remains one of the most prevalent causes of non-genetic hearing loss and developmental delay, affecting 0.5–2% of all live births globally [27,28]. In our follow-up of women with CMV infection, one case of stillbirth was documented.

One limitation of our study is the small number of patients screened, as well as the missing information on pregnancy outcomes for five of the pregnant women confirmed to have B19V and CMV infections.

Molecular methods, such as PCR assay and NGS, have shown greater sensitivity and specificity compared to traditional serology. Detection of viral DNA in maternal plasma or amniotic fluid confirms active infection and can be integrated into prenatal screening programs [29]. These results reinforce the need for proactive maternal monitoring, especially in populations with high exposure risk or limited access to serological testing.

## 4. Materials and Methods

### 4.1. Patients and Study Design

Over a twelve-month period, from April 2024 to March 2025, a total of 17 clinical samples from 13 women with pathological pregnancies, particularly those with fetal anemia, effusions, and intrauterine growth restriction, were collected at the prenatal clinic of Dr. Shterev Hospital in Sofia and were studied prospectively. The patients’ ages ranged from 19 to 41 years, with a median age of 31 ± 5.45 years. Women who were tested were between 19 and 30 gestational weeks (10 in second and 3 in third gestational trimester) pregnant. For the purposes of the study, clinical materials from pregnant women: maternal blood (n = 7) and amniotic fluid (n = 5), and from the fetus, umbilical cord serum (n = 5), were examined. These specimens were analyzed at the National Reference Laboratory for Measles, Mumps, and Rubella, which is part of the National Centre of Infectious and Parasitic Diseases in Sofia, Bulgaria, as well as at the Medical Complex “Dr. Shterev” in Sofia.

### 4.2. Enzyme-Linked Immunosorbent Assay (ELISA)

Blood samples collected from the mothers and from baby-in-uterus were tested for the presence of anti-CMV/B19V immunoglobuline G (IgG) and M (IgM) antibodies using Euroimmun ELISA kits (EUROIMMUN Medizinische Labordiagnostika AG, Lübeck, Germany).

Positive, negative, and cut-off controls were included in all runs, and results were and the results were interpreted qualitatively as positive, negative, and borderline, according to the manufacturer’s instructions.

### 4.3. DNA Extraction and Real-Time PCR for Viral Amplification

Viral DNA was extracted from all specimens using the PureLink Viral RNA/DNA Mini Kit (Thermo Fisher Scientific Inc., Waltham, MA, USA). Real-time polymerase chain reaction (qPCR) by ViroReal Kit Parvovirus B19, Ingenetix GmbH, Vienna, Austria, and CMV REAL-TIME PCR Detection Kit, “DNA-Technology Research & Production”, Moscow Region, Russia, for CMV were used. Positive and negative controls were included in each real-time PCR run. Samples testing positive for B19V with a Ct value below 30 were selected for NGS.

### 4.4. Sequencing and Phylogenetic Analysis

Next-Generation Sequencing

The targeted NGS method utilizing the Viral Surveillance Panel v2 Kit was employed to simultaneously isolate the genomes of viruses involved in mixed infections. This kit, developed by Illumina in San Diego, CA, USA, was used to characterize over 200 different viruses. NGS was conducted using the Illumina MiSeq system, Illumina, Inc., San Diego, CA, USA equipped with the 600-cycle v3 reagent kit.

Following sequencing, DNA libraries were analyzed for fragment size distribution with the QIAxcel Advanced capillary electrophoresis system (Qiagen Hilden, Germany). Library normalization was performed using the Qubit 4 Fluorometer along with the Invitrogen™ Quant-iT™ 1X High-Sensitivity (HS) Broad-Range (BR) dsDNA Assay Kit (Thermo Fisher Scientific in Waltham, MA, USA).

Genomic and phylogenetic analyses

We used the DRAGEN Microbial Enrichment Plus (DME+) software (version 1.1.1), available on the BaseSpace platform from Illumina (Cambridge, UK), for sequence assembly and FASTA file extension extraction. The B19V genetic sequences have been deposited in the GenBank sequence databases (accession numbers: PX647871, PX647872, PX647873, PX647874, PX647875, PX647876 and PX647877). BLAST searches were performed using the online NCBI BLAST tool (BLAST+ version 2.14.1) in multiple databases to retrieve references and closely related sequences. We used Geneious Prime 10.6.1 (GraphPad Software, LLC, Boston, MA, USA) for alignment, while the MEGA11 (Molecular Evolutionary Genetics Analysis) software, developed in the United States at Pennsylvania State University, was used to construct the phylogenetic tree and its overall design.

### 4.5. Statistical Analysis

Statistical analyses were performed using Microsoft Excel (Microsoft Corporation, Redmond, WA, USA):-Calculation of relative share indicators (%) to evaluate the relationship between diagnostic approaches and clinical materials used.-Standard deviation (SD) of patients’ age.

## 5. Conclusions

The findings of this study highlight the significant contribution of B19V and CMV infections during pregnancy, particularly in cases complicated by fetal anemia, effusions, and intrauterine growth restriction. A considerable proportion of the examined pregnant women showed evidence of active or recent infection, with B19V detected more frequently than CMV. This is related to the reported increased B19V circulation in 2024 in Bulgaria and Europe. The identification of B19V genotype 1a among positive samples is consistent with its known global circulation. These results support the potential benefit of implementing routine or targeted screening strategies for B19V and CMV during pregnancy follow-up, especially in high-risk cases, to enable early detection, appropriate clinical management, and improved pregnancy outcomes.

## Figures and Tables

**Figure 1 ijms-27-00427-f001:**
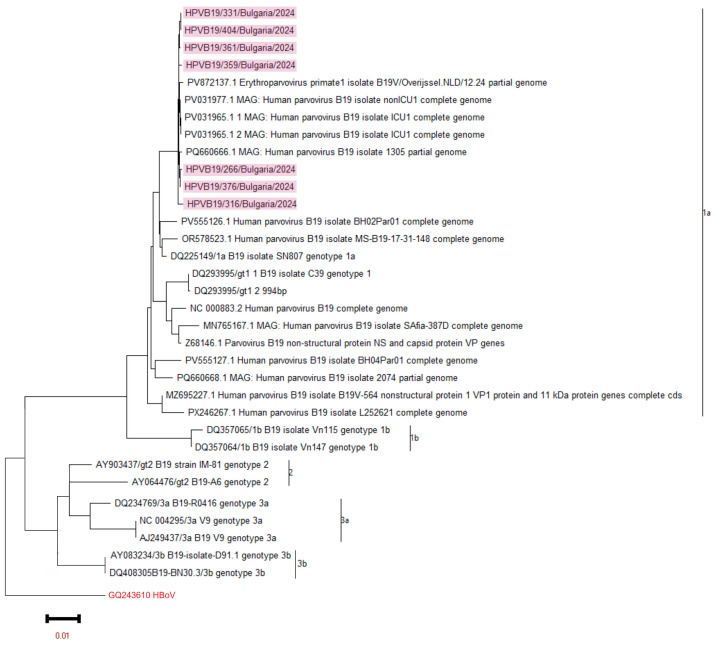
A phylogenetic analysis of B19V was conducted, focusing on the NC-VP1u protein and the complete viral genome. The phylogeny tree was generated by the ML method with 1000 bootstrap replicates. The phylogenetic tree was constructed based on these measurements. Sequences from reference strains that represent known genotypes were obtained from GenBank, along with their corresponding identification numbers. Bulgarian sequences were highlighted in blue. The tree was rooted using the strain GQ243610 HBoV as a reference point.

**Table 1 ijms-27-00427-t001:** Clinical and laboratory data of patients confirmed for B19V infection.

Cases	Age	Type of Clinical Material Tested	Clinical Complications	Laboratory Data		Pregnancy Outcome	Gestational Trimester
				ELISA	PCR		
Case 1	32	amniotic fluid	Mild pleural effusion—subsequently completely resorbed	NT	B19V DNA (+)	Live birth	Second
Case 2	41	amniotic fluid	Multiple pregnancy—Increased blood flow velocity in the middle cerebral artery, presence of ascites, and anemia in the second fetus.	NT	B19V DNA (+)	ND	Second
Case 3	35	amniotic fluid	Fetal anemia, cardiomegaly, ascites.	NT	B19V DNA (+)	ND	Second
Case 4	30	maternal blood	Severe ascites, pericardial effusion, and fetal anemia	B19V IgM (+) B19V IgG (+)	B19V DNA (+)	Normal fetal growth	Second
	baby-in-uterus	umbilical cord serum		B19V IgM (+) B19V IgG (+)	B19V DNA (+)		
Case 5	35	maternal blood	Fetal hydrops, enlarged placenta, severe pericardial effusion, severe ascites. One week after repeated intrauterine transfusion—normal fetal growth and amniotic fluid, no evidence of anemia.	B19V IgM (+) B19V IgG (+)	B19V DNA (+)	Normal fetal growth	Second
	baby-in-uterus	umbilical cord serum		B19V IgM (+) B19V IgG (+)	B19V DNA (+)		
Case 6	30	maternal blood	Mild ascites, echogenic bowel, pericardial effusion, upper limit values of amniotic fluid, severe anemia. After intrauterine blood transfusion in the second trimester, normal fetal growth and amniotic fluid were found.	B19V IgM (+) B19V IgG (+)	B19V DNA (+)	Normal fetal growth	Second
	baby-in-uterus	umbilical cord serum		B19V IgM (+) B19V IgG (+)	B19V DNA (+)		
Case 7	34	maternal blood	Ascites	B19V IgM (+) B19V IgG (+)	B19V DNA (+)	Normal fetal growth	Second
	baby-in-uterus	umbilical cord serum					

NT—Not tested; ND—No data.

**Table 2 ijms-27-00427-t002:** Clinical and laboratory data of patients confirmed for CMV infection.

Cases	Age	Type of Clinical Material Tested	Clinical Complications	Laboratory Data		Pregnancy Outcome	Gestational Week
				ELISA	PCR		
Case 1	32	amniotic fluid	Fetal growth restriction—changes in the structures of the brain, heart, and intestines	NT	CMV DNA (+)	Stillbirth	Second
				NT			
Case 2	19	umbilical cord serum	Enlargement of the fetal heart, spleen, and placenta, fetal anemia.	CMV IgM (+)	CMV DNA (+)	ND	Second
				CMV IgG (+)			
Case 3	28	amniotic fluid	Enlargement of the fetal brain ventricles	NT	CMV DNA (+)	ND	Second
				NT			

NT—Not tested; ND—No data.

## Data Availability

Data is unavailable due to privacy or ethical restrictions.

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
