# Peer review of "Congenital Viral Infection Risk: The Role of Parvovirus B19 and Cytomegalovirus Molecular Genetic Testing"

_ijms, 2025, doi:10.3390/ijms27010427_

Round 1

Reviewer 1 Report

Comments and Suggestions for Authors

This manuscript by Krumova et al is a well written manuscript about prenatal screening of congenital infections, in particular Parvovirus B19, in the settings of fetal anemia. However screening pregnant women for B19V is overall standard of care in many countries in the setting of fetal anemia/hydrops and it is not a novel management. Of course introducing this screening as part of routine care in Bulgaria could have a clinical impact and significantly reduce morbidity.

Some comments:

  • the introduction is particularly long and could consider shortening it
  • In the abstract it is stated that the indication for screening and testing the pregnany women was fetal anemia/effusions; this is not reported in the Methods and indeed the CMV positive cases (Table 2) do not support that there was fetal anemia, except in Case 2). Authors should reconcile what exactly was the criteria to screen and in case remove the CMV section.
  • in the Result section: lines 106 to 109 (These pathogens.....vaginal secretions) are redundant and are already reported in the Introduction. I would also move the lines 108 to111 (Both viruses.....morbidity) to the Introduction as it fits better in that section, rather than in the Results.
  • interesting the phylogenetic analysis of the B19V more from an epidemiological perspective rather than a management perspective.
  • in the CMV infection section: the 4th case cannot be considered a confirmed case of congenital CMV, especially using only serology. IgM can persist for a long period of time (even 1 year or more); furthermore the lack of amniotic fluid testing, the lack of pregnancy outcome and the presence of only enlargement of the fetal brain ventricles (which is a finding that can be found in many other pathologies) cannot support the diagnosis of congenital CMV. 

Author Response

Comments 1: The introduction is particularly long and could consider shortening it.

Response 1: Thank you for pointing this out. The introduction is shortened in the part about general information about viral infections and their impact on pregnancy.

Comments 2: In the abstract it is stated that the indication for screening and testing the pregnany women was fetal anemia/effusions; this is not reported in the Methods and indeed the CMV positive cases (Table 2) do not support that there was fetal anemia, except in Case 2). Authors should reconcile what exactly was the criteria to screen and in case remove the CMV section.

Response 2: Thank you for pointing this out. We have, accordingly, revised the sections Abstract (lines 28-30) and Methods and added information about the study design and patient selection (lines 211-219). Thank you for your recommendation, but we could not remove the CMV section, since the CMV study is the mainstay of laboratory screening of women with pathological pregnancies in Bulgaria.

Comments 3: In the Result section: lines 106 to 109 (These pathogens.....vaginal secretions) are redundant and are already reported in the Introduction. I would also move the lines 108 to111 (Both viruses.....morbidity) to the Introduction as it fits better in that section, rather than in the Results.

Response 3: We agree, and we moved lines 106-111 from the Results chapter to the Introduction chapter (lines 54-59).

Comments 4: Interesting the phylogenetic analysis of the B19V more from an epidemiological perspective rather than a management perspective.

Response 4: Thank you for your comment. Analyzing the phylogeny of B19V is crucial for tracking its epidemiology, particularly during epidemic outbreaks.

Comments 5: In the CMV infection section: the 4th case cannot be considered a confirmed case of congenital CMV, especially using only serology. IgM can persist for a long period of time (even 1 year or more); furthermore the lack of amniotic fluid testing, the lack of pregnancy outcome and the presence of only enlargement of the fetal brain ventricles (which is a finding that can be found in many other pathologies) cannot support the diagnosis of congenital CMV. 

Response 5: Thank you for your comment. We agree and have rethought Table 2 and some of the results section CMV.

Reviewer 2 Report

Comments and Suggestions for Authors

Lines 60-62: As written, it seems that B19V and CMV need to be co-infecting the pregnant woman to cause these clinical manifestations; it would be better to detail what each one causes.

Line 65: According to the ICTV, family and gender should be written in italics.

Line 78: Beta-herpesvirus group? This is not the correct way to represent a virus; check the subfamily and gender in the ICTV.

Line 99: I think the objective "assess the prevalence" does not reflect what was done in the work, since it was not a prevalence study, but a follow-up of pregnant women. It would be better to rewrite this objective.

Lines 106-111: This text should be in the introduction and not in the results.

Line 120 (Table 1): Use * and ** only to represent statistically significant differences; for the table legend, it is better to use letters or, even better, just list abbreviations below the table without indicating where that abbreviation is located.

Regarding this table, why was maternal blood not tested in cases 1, 2, and 3? Furthermore, it would be interesting to inform why there is no information on the outcome: was it loss to follow-up of the pregnant woman in the study or did the birth not yet occur?

Line 130 (Figure 1): Why was NGS performed on the complete B19V genome for viral genotyping? Since a Sanger sequencing methodology of the VP1-VP2 region would be sufficient for this purpose. I believe you have a lot of information about these sequences; I hope that further analyses will be done, even in a future study.

Regarding this figure, why was Neighbor-joining used instead of maximum likelihood? What method was used (HKY, T92...)? Was bootstrapping done to confirm this tree? I suggest using the HBoV sequence (GQ243610) to root the tree.

Line 147 (Table 2): As I said for Table 1, it applies here. Why wasn't ELISA performed in case 2 and PCR in case 4? Please provide details. I have the same question about the outcome of these pregnant women.

Regarding the two tables, I missed information about the gestational week/trimester when the infections occurred in both. Please include it.

Lines 150-151: Even with a high Ct (above 30), is it possible to assume that CMV infection was the cause of the pathology? I would like a discussion on this subject. And regarding sequencing, perhaps performing a nested PCR and Sanger sequencing would be possible, this has already happened to me. I think it's important to emphasize that NGS for B19V and CMV together is complicated, given the varying sizes of the genomes of these viruses.

Lines 204-206: It was mentioned that viral load may be correlated with the severity of fetal disease, but no viral load was presented in this study, therefore it is difficult to make assessments regarding this matter.

Line 210: I missed some topics in the discussion. I was missing the study's limitations, but then I saw that they were in lines 271-273, after the conclusion. I think the ideal place for these limitations would be here in the discussion.

Lines 213-221: Review this paragraph, as the information is truncated and it seems to have two sets of samples from different locations. In addition, gestational age information should be included.

Line 225: Put the manufacturer and location in parentheses.

Lines 230-237: Remove the second r from Thermo Fisher; According to the MIQE guidelines (10.1373/clinchem.2008.112797), RT-PCR means reverse transcription-preceded PCR and not real-time PCR (which should be written as qPCR); PCR machine is not the correct way to write it.

Line 253: Change PVB19 to B19V here, which is the correct form and as it appears throughout the text. Also, include the accession numbers of the sequences that were deposited in GenBank.

Lines 259-263: This section needs improvement. What was standard deviation used for? What program was used for the analyses? What method?

Lines 265-270: What was the conclusion of the study? The text in this section is about the importance of diagnosing B19V and CMV, which can be kept, provided that the conclusions of your study are included.

Lines 271-273: The limitations should not be addressed here, but in the discussion as I mentioned earlier.

Author Response

Comments 1: Lines 60-62: As written, it seems that B19V and CMV need to be co-infecting the pregnant woman to cause these clinical manifestations; it would be better to detail what each one causes.

Response 1: Thank you for your comment. The introduction chapter has been reworked, with information on clinical complications and seroprevalence of B19V (Line 60-70) and CMV (Lines 81-80) separated.

Comments 2: Line 65: According to the ICTV, family and gender should be written in italics.

Response 2: Agree. Corrections were made using Track Changes.

Comments 3: Line 78: Beta-herpesvirus group? This is not the correct way to represent a virus; check the subfamily and gender in the ICTV.

Response 3: Agree. Corrections were made by Track Changes (Lines 71-72). “CMV is a double-stranded DNA virus that belongs to the family Orthoherpesviridae, (subfamily Betaherpesvirinae, genus Cytomegalovirus)”

Comments 4: Line 99: I think the objective "assess the prevalence" does not reflect what was done in the work, since it was not a prevalence study, but a follow-up of pregnant women. It would be better to rewrite this objective.

Response 4: Thank you for your comment. The objective "assess the prevalence" was corrected in lines 92-93.

Comments 5: Lines 106-111: This text should be in the introduction and not in the results

Response 5: We agree, and we moved lines 106-111 from the Results chapter to the Introduction chapter (lines 53-59).

Comments 6: Line 120 (Table 1): Use * and ** only to represent statistically significant differences; for the table legend, it is better to use letters or, even better, just list abbreviations below the table without indicating where that abbreviation is located.

Regarding this table, why was maternal blood not tested in cases 1, 2, and 3? Furthermore, it would be interesting to inform why there is no information on the outcome: was it loss to follow-up of the pregnant woman in the study or did the birth not yet occur?

Response 6: Thank you for your comments. * and ** were removed.

Regarding the other comment, the gynecological clinic (Dr. Shterev Hospital), where the patients included in the study were followed up, is one of the largest in the country. It performs in vitro fertilization and treatment of fertility problems. The clinicians, co-authors of the manuscript, are specialists who perform interventions in pathological pregnancies. For this reason, they are a sought-after team by patients from all over the country. During our joint work, they prospectively studied these patients and included the B19V study in view of our joint scientific work and clarification of the etiology of the anemic syndrome that occurred. Primary practice in the country is the application of amniocentesis or fetal blood testing. For the purposes of the study, the treating doctors were able to provide maternal blood from four cases. The blood samples from the rest were used up.

As for the follow-up of the patients, they sought urgent medical consultation at the prenatal clinic of Dr. Shterev Hospital in Sofia and, after clarification of their condition and/or improvement, continued their follow-up with their treating gynecologist.

We added more information – lines 115-119.

Comments 7: Line 130 (Figure 1): Why was NGS performed on the complete B19V genome for viral genotyping? Since a Sanger sequencing methodology of the VP1-VP2 region would be sufficient for this purpose. I believe you have a lot of information about these sequences; I hope that further analyses will be done, even in a future study.

Response 7: For our current project, we are using NGS sequencing for a group of viruses. The method used allows for the simultaneous sequencing of 250 viruses. Our goal was to identify co-infections with other viruses, but because the concomitant viruses had high viral loads, we were unable to obtain sufficiently complete sequences, including those of CMV. These sequences will be analyzed in our other studies to identify mutations by amino acid analysis.

Comments 8: Regarding this figure, why was Neighbor-joining used instead of maximum likelihood? What method was used (HKY, T92...)? Was bootstrapping done to confirm this tree? I suggest using the HBoV sequence (GQ243610) to root the tree.

Response 8: Thank you for your comments. We have corrected the information for Figure 1 with: “A phylogenetic analysis of B19V was conducted, focusing on the NC-VP1u protein and the complete viral genome. The phylogeny tree was generated by the ML method with 1000 bootstrap replicates. The phylogenetic tree was constructed based on these measurements. Sequences from reference strains that represent known genotypes were obtained from GenBank, along with their corresponding identification numbers. Bulgarian sequences were highlighted in blue. The tree was rooted using the strain GQ243610 HBoV as a reference point.”

As well as the tree was rooted using the strain GQ243610 HBoV, according to your recommendations. 

Comments 9: Line 147 (Table 2): As I said for Table 1, it applies here. Why wasn't ELISA performed in case 2 and PCR in case 4? Please provide details. I have the same question about the outcome of these pregnant women.

Response 9: We agree with your comments. For the purposes of the study, primary fetal blood or amniotic fluid was collected to conduct PCR testing for molecular detection of B19V and CMV.

For the purposes of the study, the treating doctors were able to provide maternal blood from one CMV case. The blood samples from the rest were spent and insufficient.

Regarding the follow-up of the patients, they sought urgent medical consultation at the prenatal clinic of Dr. Shterev Hospital in Sofia. After clarification of their condition and/or improvement, they continued their follow-up with their treating gynecologist.

Regarding case 2, an additional ELISA test was performed; the results are added to Table 2.

Comments 10: Regarding the two tables, I missed information about the gestational week/trimester when the infections occurred in both. Please include it.

Response 10:  Thank you for your comment. The information on gestational age (trimester) has been added in an additional column at the end of the Table 1 and 2.

Comments 11: Lines 150-151: Even with a high Ct (above 30), is it possible to assume that CMV infection was the cause of the pathology? I would like a discussion on this subject. And regarding sequencing, perhaps performing a nested PCR and Sanger sequencing would be possible, this has already happened to me. I think it's important to emphasize that NGS for B19V and CMV together is complicated, given the varying sizes of the genomes of these viruses.

Response 11: Thank you for your comment. As we explained in the answer above, “For our current project, we are using NGS sequencing for a group of viruses. The method used allows for the simultaneous sequencing of 250 viruses. Our goal was to identify co-infections with other viruses, but because the concomitant viruses had high viral loads, we were unable to obtain sufficiently complete sequences, including those of CMV. These sequences will be analyzed in our other studies to identify mutations by amino acid analysis.”

A later cycle of positivity (above 30) can be an indication of a passing infection; all controls and validations of the test were met, and according to the test instructions, samples with a Ct before the 40th cycle are considered positive.

Comments 12: Lines 204-206: It was mentioned that viral load may be correlated with the severity of fetal disease, but no viral load was presented in this study, therefore it is difficult to make assessments regarding this matter.

Response 12: We agree with your comments. The paragraph was corrected, in the part comment on quantitative PCR results (lines 203-205).

Comments 13: Line 210: I missed some topics in the discussion. I was missing the study's limitations, but then I saw that they were in lines 271-273, after the conclusion. I think the ideal place for these limitations would be here in the discussion.

Response 13: Thank you for your recommendation. The study's limitations have been moved to the discussion section (lines 199-201)

Comments 14: Lines 213-221: Review this paragraph, as the information is truncated and it seems to have two sets of samples from different locations. In addition, gestational age information should be included.

Response 14: We agree with your comments. The paragraph was corrected, and information for gestational age was included (lines 211-219).

Comments 15: Line 225: Put the manufacturer and location in parentheses.

Response 15: EUROIMMUN Medizinische Labordiagnostika AG, Lübeck, Germany were included.

Comments 16: Lines 230-237: Remove the second r from Thermo Fisher; According to the MIQE guidelines (10.1373/clinchem.2008.112797), RT-PCR means reverse transcription-preceded PCR and not real-time PCR (which should be written as qPCR); PCR machine is not the correct way to write it.

Response 16: Thank you for your comments. The corrections have been made.

Comments 17: Line 253: Change PVB19 to B19V here, which is the correct form and as it appears throughout the text. Also, include the accession numbers of the sequences that were deposited in GenBank.

Response 17: Thank you for your comments. The corrections have been made and accession numbers (PX647871, PX647872, PX647873, PX647874, PX647875, PX647876 and PX647877) of the sequences were included.

Comments 18: Lines 259-263: This section needs improvement. What was standard deviation used for? What program was used for the analyses? What method?

Response 18: Thank you for your comments. Information was included lines 262-266.

Comments 19: Lines 265-270: What was the conclusion of the study? The text in this section is about the importance of diagnosing B19V and CMV, which can be kept, provided that the conclusions of your study are included.

Response 19: Thank you for your comments. The conclusion was corrected (lines 268-277)

Comments 20: Lines 271-273: The limitations should not be addressed here, but in the discussion as I mentioned earlier.

Response 20: The study's limitations have been moved to the discussion section (lines 199-201)

Reviewer 3 Report

Comments and Suggestions for Authors

Dear authors,

thank you for the opportunity to review your manuscript “Congenital Viral Infection Risk: The Role of Parvovirus B19 and Cytomegalovirus Molecular Genetic Testing” examining the need for screening for Parvovirus B19 and Cytomegalovirus in high-risk pregnancies. This is an important and clinically relevant topic, and your study highlights that 11 of the 13 included pregnancies showed evidence of active infection with either CMV or Parvovirus B19. The focus of your work is timely and has the potential to contribute valuable insights to the field. However, several aspects of the manuscript would benefit from clarification and further development to strengthen its scientific relevance. In its current form, the conclusions drawn do not appear to be fully supported by the presented data. Additionally, the methodology and results sections contain important ambiguities. I encourage you to revise the manuscript with the points listed in the attached document. 

Author Response

Major issues: 1) Page 1, line 40-41: The conclusion drawn by the authors is not reasonable and cannot be derived from the study results. The only conclusion that could be interpreted from the presented data might be: “Women with pregnancies complicated by fetal anemia and/or effusion are likely to be diagnosed with Parvovirus B19 or CMV.” The authors should reevaluate and revise their conclusion accordingly. Additionally, another possible conclusion could be that screening for Parvovirus B19 and CMV is recommended in such high-risk pregnancies. However, this is not a new finding, and therefore the novelty of the study’s results is questionable.

Response 1: Thank you for your comments. The authors corrected the manuscript in the abstract and conclusion section.

2) Page 3, line 104: 13 patients is a very low number, calculations and conclusions might not be applicable. Did authors include all patients with fetal anemia and/or fetal effusion in the given time period?

Response 2: The authors included all patients with fetal anemia and effusions who sought specialized help at the prenatal clinic of Dr. Shterev Hospital in Sofia during the period. There were total 17, of which 11 had acute infection with B19 or CMV. In the remaining 2, neither of the two viruses that are the focus of this study were proven (serologically or molecularly).

3) Page 7, line 211: The Methods section lacks several key elements, including a clear description of the study design (were patients included prospectively or retrospectively?) and the specific inclusion criteria. The latter appears inconsistent throughout the manuscript. For example, the abstract states that the inclusion criteria were “pregnant women with fetal anemia and/or effusion”, yet in Table 2 several cases (e.g., case 1, case 3, and case 4) do not present with fetal anemia and/or effusion. The authors should clarify these discrepancies. In addition, more detailed information on data collection is needed. It is unclear whether all patients underwent serological testing and amniocentesis. If amniocentesis was performed, was this due to fetal anomalies, or because maternal serology suggested a congenital infection? Furthermore, clarification is required regarding case 2 in Table 2—why was fetal blood collected? The description of umbilical cord blood sampling is also ambiguous. Was cord blood obtained postnatally? The phrase “baby in utero” may be misleading if samples were actually collected after delivery. However, testing for congenital infections is already recommended in pregnancies complicated by fetal anemia. The manuscript would benefit from additional context regarding current clinical guidelines and routine practice in Bulgaria. Are there national recommendations for screening for congenital infections in high-risk pregnancies? Providing this information would help readers better understand the clinical framework within which the study was conducted. (see Khalil A, Sotiriadis A, Chaoui R, da Silva Costa F, D'Antonio F, Heath PT, Jones C, Malinger G, Odibo A, Prefumo F, Salomon LJ, Wood S, Ville Y. ISUOG Practice Guidelines: role of ultrasound in congenital infection. Ultrasound Obstet Gynecol. 2020 Jul;56(1):128-151. doi: 10.1002/uog.21991. Epub 2020 May 13. PMID: 32400006.)

Response 3: Thank you for your comments. According to your guidelines, corrections have been made to the manuscript in the chapter Materials and Methods (study design), as well as in the chapter Results (Tables 1 and 2). The authors would like to emphasize that the study is prospective; in Bulgaria, there is no established standard for the study of cases with pathological pregnancy in relation to B19. It is carried out in some cases, as part of the TORCH syndrome study. In the course of the study, clinical materials from pregnant women were examined: maternal blood (n=7) and amniotic fluid (n=5), and from the fetus, umbilical cord serum (n=5). Also, all fetal samples, including blood transfusions to control anemia, were taken in utero, not after birth.

4) Page 3, line 120 (table 1): The outcome parameters should be defined more uniformly. For example, a pregnancy with normal fetal growth may still result in a live birth, while the outcome “live birth” alone does not provide any information about the fetal or neonatal condition, such as growth parameters or APGAR scores. To improve clarity and consistency, the authors should select and apply uniform outcome measures across all cases.

Response 4: Thank you for your comments. According to your guidelines, corrections have been made to the manuscript in the chapter Results.

5) Page 5, line 154: The Discussion section is currently very poorly structured and consists primarily of general background information. A critical analysis and interpretation of the study’s own results are largely missing. The authors should restructure the Discussion to focus more clearly on their findings, place them in the context of existing literature, and address the strengths and limitations of their study.

Response 5: Thank you for your comments. According to your guidelines, corrections have been made to the manuscript in the chapter Discussion.

Minor issues: 1) Page 7, line 211: Information about the gestational age at time of testing/diagnosis would be more relevant than the maternal age. The authors should add the gestational age of included patients.

Response 1: Thank you for your comments. Information about the gestational age was included in the chapter Results and Tables 1 and 2

2) Page 8, line 271-271: Limitations of the study should be part of the discussion section. The authors should correct this.

Response 2: The study's limitations have been moved to the discussion section (lines 199-201)

3) Page 4, line 123-125: It is unclear whether intrauterine blood transfusion was performed in all Parvovirus B19 cases or only in those with severe anemia, in accordance with international guidelines. Additionally, information on the number of transfusions performed and the gestational age at which they were conducted would be highly relevant. The authors should provide these details to improve clarity and allow better interpretation of the management strategies used.

Response 3: The information was included. We would like to emphasize that the study is focused on B19V and CMV detection during pregnancy and in certain high-risk groups of patients, in view of the widespread presence of the B19V virus in Bulgaria and Europe, and not with a clinical focus.

4) Page 4, line 127: Abbreviations such as “NGS” should be clearly defined at first use to ensure clarity for the reader.

Response 4: Thank you for your comments. Abbreviations was defined on line 91.

5) Page 7, line 216: “notably, only two woman were younger than 30” does not add meaningful value to the manuscript and appears irrelevant to the reader. It may be removed unless a clear clinical relevance is demonstrated.

Response 5: Thank you for your comments. It was removed. 

Round 2

Reviewer 1 Report

Comments and Suggestions for Authors

Manuscript was modified according to initial reviewer suggestions and is more readable. In the Conclusions authors eliminated the limitation of the small number, which overall is a true statement. Consider reintroducing that paragraph, but this should not compromise acceptance

Reviewer 2 Report

Comments and Suggestions for Authors

Since all suggestions have been addressed, I believe the article can proceed to publication, as it provides important information about B19V and CMV infection in pregnant women, an important topic for this population.